# Dual Effect of Organogermanium Compound THGP on RIG-I-Mediated Viral Sensing and Viral Replication during Influenza a Virus Infection

**DOI:** 10.3390/v13091674

**Published:** 2021-08-24

**Authors:** Sunanda Baidya, Yoko Nishimoto, Seiichi Sato, Yasuhiro Shimada, Nozomi Sakurai, Hirotaka Nonaka, Koki Noguchi, Mizuki Kido, Satoshi Tadano, Kozo Ishikawa, Kai Li, Aoi Okubo, Taisho Yamada, Yasuko Orba, Michihito Sasaki, Hirofumi Sawa, Hiroko Miyamoto, Ayato Takada, Takashi Nakamura, Akinori Takaoka

**Affiliations:** 1Division of Signaling in Cancer and Immunology, Institute for Genetic Medicine, Hokkaido University, Sapporo 060-0815, Japan; sunanda@igm.hokudai.ac.jp (S.B.); wanchiki4100@live.jp (Y.N.); sakurain@igm.hokudai.ac.jp (N.S.); nonaka@igm.hokudai.ac.jp (H.N.); nk810ru@igm.hokudai.ac.jp (K.N.); utapon1202@igm.hokudai.ac.jp (M.K.); s.tadano@frontier.hokudai.ac.jp (S.T.); oup3p.37725126c@gmail.com (K.I.); likai.19870816@163.com (K.L.); aoringo621@igm.hokudai.ac.jp (A.O.); t-yamada@igm.hokudai.ac.jp (T.Y.); 2Molecular Medical Biochemistry Unit, Biological Chemistry and Engineering Course, Graduate School of Chemical Sciences and Engineering, Hokkaido University, Sapporo 060-0815, Japan; 3Asai Germanium Research Institute Co., Ltd. Suzuranoka, Hakodate 042-0958, Japan; y.shimada@asai-ge.co.jp (Y.S.); nakamura@asai-ge.co.jp (T.N.); 4Division of Molecular Pathobiology, International Institute for Zoonosis Control, Hokkaido University, Sapporo 001-0020, Japan; orbay@czc.hokudai.ac.jp (Y.O.); m-sasaki@czc.hokudai.ac.jp (M.S.); sawa@czc.hokudai.ac.jp (H.S.); 5International Collaboration Unit, International Institute for Zoonosis Control, Hokkaido University, Sapporo 001-0020, Japan; atakada@czc.hokudai.ac.jp; 6Global Virus Network, Baltimore, MD 21201, USA; 7Division of Global Epidemiology, International Institute for Zoonosis Control, Hokkaido University, Sapporo 001-0020, Japan; hirom@czc.hokudai.ac.jp

**Keywords:** influenza a virus, viral replication, recognition of 5′-triphosphate RNA, antiviral agent, THGP, RIG-I

## Abstract

The interaction of viral nucleic acid with protein factors is a crucial process for initiating viral polymerase-mediated viral genome replication while activating pattern recognition receptor (PRR)-mediated innate immune responses. It has previously been reported that a hydrolysate of Ge-132, 3-(trihydroxygermyl) propanoic acid (THGP), shows a modulatory effect on microbial infections, inflammation, and immune responses. However, the detailed mechanism by which THGP can modify these processes during viral infections remained unknown. Here, we show that THGP can specifically downregulate type I interferon (IFN) production in response to stimulation with a cytosolic RNA sensor RIG-I ligand 5′-triphosphate RNA (3pRNA) but not double-stranded RNA, DNA, or lipopolysaccharide. Consistently, treatment with THGP resulted in the dose-dependent suppression of type I IFN induction upon infections with influenza virus (IAV) and vesicular stomatitis virus, which are known to be mainly sensed by RIG-I. Mechanistically, THGP directly binds to the 5′-triphosphate moiety of viral RNA and competes with RIG-I-mediated recognition. Furthermore, we found that THGP can directly counteract the replication of IAV but not EMCV (encephalitismyocarditis virus), by inhibiting the interaction of viral polymerase with RNA genome. Finally, IAV RNA levels were significantly reduced in the lung tissues of THGP-treated mice when compared with untreated mice. These results suggest a possible therapeutic implication of THGP and show direct antiviral action, together with the suppressive activity of innate inflammation.

## 1. Introduction

The innate immune system is the first line of host defense against invasion by a variety of microbes. The molecular sensor-mediated recognition of invading microbes is the first critical step that initiates the activation of this system. In particular, during viral infections, virus-associated molecular patterns, viral nucleic acids (RNA and DNA), are mainly targeted by pattern recognition receptors (PRRs), including transmembrane-type Toll-like receptors (e.g., TLR3 and TLR9) and cytoplasmic sensors, such as RIG-I (retinoic acid-inducible gene I) and cGAS (cyclic GMP-AMP synthetase) [1,2,3,4]. In most cases, such viral sensors activate their downstream signaling to induce types I and III IFN genes, which confer antiviral states to the cell. Among viral nucleic acid sensors, RIG-I and MDA5 are ubiquitously expressed and recognize viral RNAs in the cytoplasmic space. Particularly, RIG-I is a key cytoplasmic PRR for the detection of RNA viruses, such as influenza virus, hepatitis C virus, hepatitis B virus, measles virus, etc., that can be responsible for human infectious diseases [5,6,7,8]. RNA carrying 5’-triphosphate modification (3pRNA) and/or short dsRNA is an essential determinant for RIG-I recognition. Ligand-binding to RIG-I activates the ATPase activity to change its conformation. Then oligomerized RIG-I interacts with the adaptor protein MAVS/IPS-1 through the CARD domains, leading to the activation of the downstream gene induction programs such as IFNs and proinflammatory cytokines [1,4].

Germanium (Ge) is a trace element, and compounds of Ge are classified into organic and inorganic forms, which are present in certain plants, such as garlic, water-nut, and pearl barley [9,10]. Poly-trans-[(2-carboxyethyl) germasesquioxane] (Ge-132) is a water-soluble organogermanium compound that is also known as bis(2-carboxyethylgermanium) sesquioxide, repagermanium, or 2-carboxyethyl-germasesquioxane. Ge-132 is a polymer and hydrolyzes to 3-(trihydroxygermyl)propanoic acid (THGP) monomer spontaneously in the presence of water. There is accumulating evidence that the hydrolytic monomer of Ge-132, THGP (Appendix A for the chemical structure) exhibits diverse biological activities, such as anti-inflammatory, anti-oxidative, anti-melanogenic, pain relief, immunostimulatory, and tumor suppressive effects [11,12,13,14,15,16,17]. Due to the low toxicity of THGP, Ge-132 is used as a dietary supplement and in cosmetics in the United States, Europe, and Japan [18,19,20,21], and much attention has been paid to this compound, in that it may have a potential therapeutic application in a wide range of fields including cancer, inflammatory diseases, and infectious diseases.

It was previously reported that this compound shows a protective effect in mice infected with a mouse-adapted strain of influenza virus (H2N2) [22], which was based on the THGP effect through IFN-γ inducing activity. However, the detailed mechanisms of the immunomodulatory activities of THGP, as well as its effect on innate sensor-mediated immune responses remains to be poorly understood.

In this study, we found that THGP can downregulate RIG-I-mediated signaling via its competitive inhibition of the interaction between RIG-I and its viral RNA ligand. It is likely that THGP specifically targets the 5′-triphosphate portion of the RIG-I ligand, which can also block the binding of IAV RNA polymerase to viral genome, leading to the suppression of viral replication. Thus, our findings suggest that THGP can show direct antiviral action together with a suppressive activity of RIG-I-mediated innate signaling.

## 2. Materials and Methods

### 2.1. Cell Culture

RAW 264.7, HEK293T, A549, and MDCK cells were purchased from ATCC. All cells were maintained in culture medium as recommended by ATCC. To generate MAVS KO A549 cells, we used pX330 vector [23], which was a gift from F. Zhang (Addgene plasmid #42230, Watertown, MA). The plasmid containing guide RNA against MAVS gene (5′-ATTGCGGCAGATATACTTAT-3′, Appendix A) was transfected, and puromycin-resistant A549 cells were cloned by limiting dilution. The genomic sequences of clones were verified by DNA sequencing, in addition, the absence of MAVS protein was also confirmed by using Western blotting. Cells were assayed by using 0.4% trypan blue uptake (Invitrogen) for measuring cell growth.

### 2.2. Viruses, Antibodies, and Reagents

IAV (A/Puerto Rico/8/1934 H1N1 strain), VSV (New Jersey strain), and EMCV were used as described previously [24]. Antibodies were used as follows: anti-RIG-I (#3743S; Cell Signaling Technology (CST), Danvers, MA, USA), anti-MDA-5 (#5321S; Cell Signaling), anti-MAVS (#3993S; Cell Signaling), anti-pTBK1 (pS172) (D52C2; CST), anti-TBK1 (EP611Y; Abcam, Cambridge, UK), anti-pIRF-3 (pS396) (4D4G; CST), anti-IRF-3 (39371; Active Motif, Carlsbad, CA, USA), anti-β-actin (AC-15; Sigma, St, Louis, MO, USA), anti-Flag (M2; Sigma), and anti-GST (B-14; Santa Cruz, Dallas, TX, USA). 3pRNA was prepared by using in vitro transcription under the control of the T7 promoter with MEGAscript (Ambion, Austin, TX, USA) as described previously [7]. PolyI:C and LPS derived from E. coli O111:B4 were obtained from GE Healthcare and Invivogen, respectively. HT-DNA, ATP, deoxy-ATP, actinomycin D, and Zanamivir were purchased from Sigma. Adenosine was purchased from Wako. THGP (Asai Germanium Research Institute, Kawasaki, Japan) was prepared as described previously [17]. Biotinylated 3pRNA and Cy5-3pRNA were prepared by using Label IT Nucleic Acid Labeling Kit, Biotin, or Label IT Nucleic Acid Labeling Kit, Cy5 (Mirus, Madison, WI, USA), respectively, according to the product protocol. As for the preparation of dephosphorylated 3pRNA, 3pRNA was treated with alkaline phosphatase (Roche, Basel, Switzerland), according to the product protocol. 

### 2.3. MAMPs Stimulation and Transfection

Stimulation with 3pRNA, polyI:C, and HT-DNA were conducted using Lipofectamine 2000 Reagent (Invitrogen, USA) according to the product protocol. The final concentration of 3pRNA, polyI:C, and HT-DNA were 1 μg/mL, 1 μg/mL, and 4 μg/mL, respectively. LPS was directly added into the culture solution at a final concentration of 100 ng/mL. To overexpress RIG-I, MAVS, TBK-1, IRF-3 CA, STING, and TRIF, the plasmids were described previously [7,25,26]. The cDNA of MDA5 was amplified by KOD plus polymerase (Toyobo, Osaka, Japan) and subcloned into pIRM-3HA vector. For plasmid transfection, Lipofectamine 2000 reagent (Invitrogen) was used for HEK293T cells transfection.

### 2.4. Viral Infection

In the case of IAV infection, the culture medium was replaced with DMEM without FBS containing 0.0005% trypsin, and PR8 was added at 1.0 multiplicity of infection (MOI). VSV was infected at an MOI of 0.1. In the case of EMCV, the culture media was changed to DMEM containing no FBS, and the EMCV was infected at an MOI of 0.1. After 1 h, FBS was added to the culture medium to a final concentration of 10%. As for the preparation of viral RNA, IAV RNAs including PR8, A/Aichi/2/68 (H3N2), and X31 (H3N2) were purified viruses from egg fluid. EMCV RNA was purified viruses from culture medium of infected IFNAR1 KO cells. SARS-CoV-2 RNA was prepared from culture medium of Vero cells infected as described [24,27]. 

### 2.5. ELISA

The levels of mouse IFN-β protein in culture medium were measured by using the VeriKine-HS Mouse IFN-β Serum ELISA kit (PBL; 42410), following the manufacturer’s protocol.

### 2.6. Plaque-Forming Assay

RAW264.7 or A549 cells were infected with IAV for 24 h, culture media were collected and serially diluted from 10-1 to 10-6. MDCK cells were infected. After 1 h infection, cells were overlaid with MEM/BactoAgar/trypsin mixture (1 × MEM (Gibco, TX, USA), 0.3% BSA, 0.28% NaHCO_3_, 1 × MEM Amino Acids Solution (Gibco), 1 × MEM Vitamine Liquid (Gibco), 2 mM L-glutamine, 1 × penicillin streptomycin solution (Sigma), 0.0005% trypsin, 0.8% BactoAgar). After 2 days, the plaque number was counted.

### 2.7. qRT-PCR

RNA was collected by using ISOGEN (Nippon gene, Tokyo Japan) and dissolved in DNase and RNase free water. DNase I (Invitrogen) was added to the extracted RNA and reacted at 25 °C for 15 min to degrade the genomic DNA. After the reaction, DNase I was inactivated by adding 1 μL of 25 mM EDTA and reacting at 65 °C for 10 min. Subsequently, RNA was reverse-transcribed using ReverTra Ace qPCR RT Kit (Toyobo) to synthesize cDNA. Quantitative PCR was performed by the intercalator method using SYBR Premix Ex Taq (Takara, Shiga, Japan), quantified by the StepOnePlus Real-Time PCR System (Applied Biosystems) and analyzed by the ∆∆Ct method. Gapdh was used as an internal standard. Detailed information about the primers used here is shown in Appendix A. 

### 2.8. Immunofluorescence

RAW264.7 cells seeded on a coverslip (Matsunami Glass, Osaka, Japan) were treated with THGP for 24 h, stimulated with Cy5-3pRNA, and incubated at 37 °C for 2 h. Cells were fixed with 4% paraformaldehyde/phosphate buffer (Wako) for 20 min, permeabilized with 0.2% Triton-X for 10 min at room temperature, and blocked with 1% BSA in PBS. The coverslips were incubated for 1 h in the primary antibody diluted in 1% BSA in PBS. Following washing with PBS, the coverslips were incubated for 1 h in appropriate secondary antibodies conjugated with Alexa Fluor 488/594 (Invitrogen) diluted in 1% BSA in PBS. Subsequently, the coverslips were mounted in Slowfade Gold antifade reagent (Invitrogen) with Hoechst 33342 (Invitrogen). Confocal microscopy was performed with an IX-81S confocal microscope (Olympus, Tokyo, Japan). More than 30 cells in each condition were randomly chosen and representative images are shown in Figures.

### 2.9. FACS Analysis

The cells treated with Cy5-3pRNA in the presence of THGP were suspended in 2% FBS-PBS and transferred to a tube with a cell strainer cap (Falcon). The fluorescence in the cells was measured with FACS CantoII (BD), and a total of 20,000 cells were counted, and the proportion of cells containing the fluorescent dye was calculated.

### 2.10. Recombinant RIG-I Protein

GST-tagged RIG-I WT were expressed in Sf9 cells according to the manufacturer’s instructions for Bac-to-Bac baculovirus expression system (Invitrogen) and purified with Glutathione Sepharose 4B (GE Healthcare). GST-RIG-I protein was eluted with glutathione. The recombinant RIG-I protein was isolated to approximately >95% purity as judged by Coomassie brilliant blue staining.

### 2.11. In Vitro RNA Pull down Assay

After the incubation of indicated concentrations of THGP and biotinylated 3pRNA (100 ng) in lysis buffer (20 mM HEPES, 150 mM NaCl, 1 mM EDTA, 0.1% NP-40, 1 mM PMSF) for 30 min at room temperature, 2 μg of GST-RIG-I was added, and the solution was further mixed by inverting at room temperature for 1 h. Next, Dynabeads M-280 Streptavidin (Invitrogen) were added and mixed by inverting for 1 h at room temperature and washed 3 times with wash buffer (20 mM HEPES, 150 mM NaCl, 1 mM EDTA, 0.1% NP-40). Then, 20 μL of 2 × sample buffer was added and then boiled at 100 °C for 5 min, then the sample was subjected to Western blotting.

### 2.12. Binding Assay Using THGP Immobilized Column

For the preparation of THGP-conjugated beads, THGP was immobilized with TOYOPEARL AF-Epoxy-650 (Tosoh Bioscience, Tokyo, Japan) via ring-cleavage reaction between the epoxy group and amino acid derivative of THGP. The unreacted epoxy group was inactivated with mono-ethanolamine. As for binding assay, the beads were pretreated with 0.5 M NaOH and washed with RNase-free water until the pH reached 7.0. 3pRNA, polyI:C, or HT-DNA (2 μg each) were added to a THGP-immobilized column or control beads, the mixture was incubated with gentle rotation at room temperature for 60 min and centrifuged for 5 min. The beads were washed 3 times with 100% methanol and eluted with 0.5 M HCl. The concentration of the nucleic acids was measured using a micro ultraviolet visible spectrophotometer Q5000 (Tomy).

### 2.13. Luciferase Assay

As for the measurement of the promoter activity of IFN-β and IFN-stimulated response element (ISRE), cells seeded on 24-well plates were transiently cotransfected with luciferase reporter plasmids (100 ng each of pISRE-Luc (Clontech) and p-125Luc (provided by T. Fujita), together with an expression vector or control vector. As for the IAV luciferase activity based mini genome assay for the measurement of IAV polymerase activity, cells seeded on 24-well plates were co-transfected with expression plasmids for WSN-PB2, -PB1, -PA, and -NP (50 ng each); pPolI-NP(0)luc2(0) (2.5 ng) expresses the firefly luciferase gene between the noncoding regions of the WSN-NP gene (provided by Y. Kawaoka), as described previously [28]. As an internal control, 5 ng renilla luciferase reporter plasmid was transfected simultaneously. At 24 h after transfection, cells were stimulated with the indicated concentrations of THGP for 24 h, and luciferase activity was measured with the Dual-Luciferase Reporter Assay system (Promega) and a photon counter (Centro LB 960; BERTHOLD).

### 2.14. RIP Assay 

A549 MAVS KO cells were lysed with buffer A (20 mM HEPES, 150 mM NaCl, 1 mM EDTA, 1% NP-40, 1 mM PMSF, 1 mM DTT, 1 μg/mL leupeptin, and 100 U/mL RNaseOUT (Invitrogen) [pH 7.3]), and 20 μL of the supernatant was saved as input. Anti-Flag or control IgG was added to cell lysates. After 2 h incubation with the antibody, Protein-G Dynabeads (Invitrogen) were added, and the solution was further incubated for 1 h with gentle shaking. Beads were washed three times with a wash buffer (20 mM HEPES, 150 mM NaCl, 1 mM EDTA, 0.5% NP-40 [pH 7.3]). The immunoprecipitated IAV RNA-like RNA of the reporter gene were eluted with Isogen and analyzed by qRT-PCR with specific primers for firefly luciferase (Appendix A). The amount of immunoprecipitated RNAs is represented as a percentile of the amount of input RNA (%input).

### 2.15. Mice Experiment

MAVS KO mice (#008634) were obtained from Jackson Laboratory. C57BL/6J mice were obtained from Crea Japan. Mice were pretreated with 30 μL of THGP (50 or 100 mg/kg) by i.n. injection at 3 h before infection. Mice were infected with IAV (A/Puerto Rico/8/1934 H1N1 strain, 1 × 10^5^ pfu/mice; i.n.). During the treatment mice were anesthetized with isoflurane (Baxter). At the indicated time after infection, the lung tissues were used for further experiments. These animal experiments were approved by the committee reviews of Hokkaido University (No. 19-0041). For Hematoxylin-Eosin Stain, formalin-fixed paraffin-embedded lung tissues were stained with Mayer’s hematoxylin (Wako) and 1% Eosin Y (Wako) following the manufacturer’s protocol.

### 2.16. ^1^H-NMR

The ^1^H-NMR spectrum of each sample was measured using a Mercury Plus 300 MHz instrument (Agilent Technologies Inc., Little Falls, CA, USA) as described previously [17].

### 2.17. Quantification and Statistical Analysis

Values are shown as mean ± s.d. Statistical significance between two samples was determined with a Student’s t-test. A log–rank test was used to test for differences in survival between control and THGP-treated mice after IAV infection.

## 3. Results

### 3.1. THGP Inhibits RIG-I-Mediated IFN-β mRNA Induction 

To first examine whether THGP affects pattern recognition receptor-mediated innate immune signaling pathways, we measured IFN-β mRNA induction in a murine macrophage cell line, RAW264.7, pretreated with THGP upon stimulation with 5′-triphosphate RNA (3pRNA), polyI:C, herring testis-DNA (HT-DNA), and lipopolysaccharide (LPS), which are ligands for RIG-I, MDA5, cGAS, and TLR4, respectively (Appendix A). Treatment with THGP inhibited the IFN-β mRNA response induced by 3pRNA but not the other ligands in a dose-dependent manner (Appendix A). We also confirmed that the production of IFN-β in response to 3pRNA but not the others is suppressed by THGP at protein levels (Figure 1A–D). In addition, treatment with THGP also suppressed the phosphorylation of both TBK1 and IRF-3 induced by 3pRNA but not the other ligands (Figure 1E–H). On the other hand, stimulation with THGP alone neither induce IFN-β mRNA induction nor affect cell growth and viability (Appendix A). These results suggest that THGP has a suppressive effect on 3pRNA-induced activation of RIG-I pathway.

### 3.2. THGP Attenuates IFN-β mRNA Induction in Response to IAV and VSV Infections

Next, we investigated whether THGP affects the activation of the RIG-I pathway during viral infections. To address this, we tested the effect of THGP on cytokine responses to infections with IAV or vesicular stomatitis (VSV), which are known to be mainly detected by RIG-I. Consistent with the result shown in Figure 1A, IFN-β mRNA induction in response to IAV or VSV infection was suppressed by treatment with THGP in a dose-dependent manner (Figure 2A,B), while THGP did not affect IFN-β mRNA induction in response to infection with EMCV, which is recognized by MDA5 and does not produces triphosphate RNA [5] (Figure 2C). Similar observation was also obtained for other cytokines induced by innate sensors, such as IL-6, and TNF-α (Figure 2D–I). In addition, we confirmed that THGP reduced the mRNA inductions of IFN-β and related genes in response to 3pRNA and IAV infection in A549 cells (Appendix A). These results suggest that THGP can suppress cytokine responses downstream of RIG-I but not the MDA5 pathway upon viral infections (Appendix A). 

### 3.3. THGP Reduces the Interaction between RIG-I and Its Ligand

To clarify the mechanism for how THGP suppresses the RIG-I pathway, we tried to determine at what level THGP acts on the RIG-I signaling pathway (Appendix A). We first checked whether THGP affected the activation of the IFN-β gene promoter upon the overexpression of RIG-I or its adaptor MAVS. Interestingly, while 3pRNA-induced IFN-β gene activation was suppressed by THGP treatment in a dose-dependent manner, it failed to show such a suppressing activity upon RIG-I overexpression even at a high concentration of THGP (Figure 3A,B). In addition, similar results showed that THGP does not affect IFN-β gene activation induced by the overexpression of MDA-5, MAVS, STING, TBK1, TRIF, and a constitutively active form of IRF-3 (K152R) (Figure 3C and Appendix A). These results suggest that THGP may act on some process(es) upstream of RIG-I, such as the transfection of 3pRNA into the cytoplasm and/or its interaction with RIG-I. 

We next examined the effect of THGP on the uptake of 3pRNA into the cell. We used Cy5-labelled 3pRNA for transfection into RAW264.7 cells, which were then subjected to FACS and immunofluorescence analyses. However, no effect of THGP treatment on the uptake of Cy5-labeled 3pRNA was observed (Figure 3D and Appendix A). We further tested whether THGP interferes with the interaction between RIG-I and its ligand. Therefore, we prepared THGP-conjugated beads and examined the binding of THGP to RIG-I ligands, 3pRNA, and short-form ds-RNA. We found that THGP preferentially associated with 3pRNA but not short-form polyI:C (approximately 4 kb) or HT-DNA (Figure 3E). Such an interaction was not observed when we used dephosphorylated 3pRNA, which suggested that its 5′-triphosphate moiety was a key target for the interaction of THGP with 3pRNA (Figure 3F). On the other hand, THGP itself did not show a binding activity to RIG-I protein (Figure 3G). In addition, an RNA pull-down assay with biotin-conjugated 3pRNA showed that endogenous RIG-I was co-precipitated with biotin-conjugated 3pRNA, which was competitively inhibited by THGP treatment (Figure 3H and Appendix A). On the other hand, THGP did not affect the interaction between MDA5 and polyI:C (Figure 3I). In addition, we observed that the intracellular colocalization of RIG-I with 3pRNA in RAW264.7 cells was suppressed by THGP (Figure 3J). These results suggest that THGP binds to 3pRNA to suppress the interaction between RIG-I and 3pRNA, which leads to the suppressed activation of RIG-I pathway. 

### 3.4. THGP Impedes the Interaction of IAV Polymerase with Viral RNA Genome

We next evaluated the effect of THGP treatment on viral genome replication upon infection with IAV, which produces RNA species carrying a 5′-triphosphate moiety that can be a ligand of RIG-I. THGP treatment significantly reduced the viral titers 24 h after infection without the toxic effect of THGP (Figure 4A and Appendix A). Therefore, we speculated that THGP binding to IAV RNAs might exert a direct antiviral activity on viral replication. In order to test this hypothesis, we used the luciferase activity-based mini genome assay [29] to quantitatively evaluate IAV replication in a human lung cell line, A549. As shown in Figure 4B, we observed that IAV replication was reduced by THGP in a dose-dependent manner (Figure 4B). In addition, we used the CRISPR/Cas system to generate MAVS-deficient A549 cells, which failed to induce both RIG-I- and MDA5-mediated innate responses such as type I IFN induction (Appendix A). THGP comparatively also showed a dose-dependent suppressing effect on IAV replication in MAVS-deficient cells (Figure 4C), wherein the activation of the IFN-stimulated response element (ISRE)-driven luciferase gene was not observed (Appendix A). These results suggest that THGP has direct antiviral activity, possibly through its interaction with the 5′-triphosphate of the IAV genome. The THGP beads pull down assay showed that IAV-derived RNAs, including PR8, Aichi, and X31 but not EMCV and SARS-CoV-2-derived RNA, was remarkably co-precipitated with THGP conjugated with epoxy beads (Figure 4D). In this regard, since it is known that the 5′-region of viral genome contains one of the binding sites of IAV-derived RNA polymerase complex [30], we hypothesized that THGP abrogated the access of IAV polymerase to viral RNA. We used MAVS-deficient A549 cells to remove the possible effect of IFN or IFN-inducible proteins on the interaction between the polymerase and viral RNA-like RNA. As we expected, the RIP assay showed that THGP competitively suppressed the interaction of Flag-tagged polymerase subunit basic protein 2 (PB2) with viral RNA-like RNA in MAVS-deficient A549 cells, in which IAV polymerase subunits, PB1, PA, and NP, as well as Flag-tagged PB2, were reconstituted (Figure 4E). Furthermore, treatment with THGP restored the body weight loss and improved the survival rates of IAV-infected wild-type mice (Figure 4F,G), although the treatment with THGP alone did not show any toxic effect of THGP on survival or the body-weight curves of uninfected mice (Appendix A). Lastly, we used MAVS-deficient mice to assess the effect of THGP on IAV infection in vivo. The viral titers on Day 2 after IAV infection were significantly decreased by THGP treatment (Figure 4H). Further, hematoxylin and eosin staining showed that the administration of THGP reduced alveolar hemorrhage, inflammatory infiltration, and interstitial thickening in the alveolar lesions of lung tissue during IAV infection (Figure 4I). Thus, these data suggest that THGP is capable of negatively regulating IAV replication, possibly by blocking the interaction of IAV RNA and viral polymerase. Therefore, THGP functions as a direct antiviral agent that inhibits viral RNA recognition during IAV replication.

## 4. Discussion

It has previously been reported that an organo-germanium compound THGP, which is a hydrolysate of Ge-132, shows an antiviral effect on IAV infection. However, the detailed mechanisms of the antiviral activities of THGP, as well as its effect on innate sensor-mediated immune responses were poorly understood. Here, we represented a novel mechanism, by which THGP inhibited not only IAV replication but also RIG-I signaling. For the first time, we demonstrated the suppressing effect of an organogermanium compound, THGP, on the innate signaling pathways activated by RIG-I. In particular, 3pRNA but not ds-RNA-mediated cytokine response was selectively suppressed by THGP treatment. In addition, THGP did not affect other innate signaling mediated by MDA-5, cGAS, and TLR4. Consistent with these results, we also showed that THGP attenuated type I IFN responses and inflammatory cytokines such as IL-6 and TNF-α upon infections with VSV and IAV, which are reported to be mainly sensed by RIG-I [5,31], whereas such a suppressing effect was not observed upon infection with EMCV, which is recognized by MDA5 [32]. Interestingly, the suppressive effect was not seen by the overexpression of RIG-I and the related signaling proteins such as MDA-5, MAVS, STING, TBK-1, IRF-3, and TRIF. Further investigation revealed that THGP interacted with 3pRNA, which abrogated the RIG-I-mediated recognition of 3pRNA but not the uptake of 3pRNA. According to our data, THGP preferentially interacts with 3pRNA but not polyI:C and HT-DNA. Several reports previously showed that THGP has the characteristic of interacting with compounds containing cis-diol bonds, such as adrenaline, ATP, and L-DOPA [17,19]. Further, an ^1^H-NMR analysis comparing ATP and deoxy-ATP here indicated that cis-diol of ATP plays an important role in binding with THGP (Appendix A). In this respect, it was presumable that the cis-diol group of 3pRNA would be a binding site of THGP, but our data indicated that modified 3pRNA, which contain one cis-diol at 3′ end, can still bind to THGP (Figure 3E). The 5′ triphosphate group of 3pRNA was found to be a key portion that was targeted by THGP, which was supported by the result that the dephosphorylation of 3pRNA partially lost the binding activity with THGP (Figure 3F). In addition, our pull-down assay using THGP beads in the presence of competitors, including ATP, deoxy-ATP, or adenosine, suggests that the triphosphate region is more important rather than the cis-diol for the interaction between 3pRNA and THGP (Appendix A). A further detailed X-ray crystallographic analysis is required to structurally show the complex formation between 3pRNA and THGP. 

We also showed that THGP interacted with IAV-derived RNA but not EMCV-derived RNA or SARS-CoV-2-derived RNA. The THGP-mediated inhibition of RIG-I signaling led us to test whether THGP promoted viral replication. However, we observed that IAV replication was also suppressed by THGP treatment. Our current data indicated that THGP has a dual effect on the RIG-I-mediated IFN pathway and viral replication, which are in opposition to each other, eventually resulting in decreased viral replication. Mechanistically, we found that THGP binding to IAV RNA restricted the accessibility of viral RNA polymerase to the RNA genome to initiate viral replication without affecting viral RNA stability (Figure 4E, data not shown), suggesting a possible novel direct antiviral action of THGP on IAV RNA or other virus-derived RNAs containing a 5′-triphosphate moiety. It should be noted that viral RNAs of EMCV and SARS-CoV-2, which contains 7-methylguanosine (m7G) cap at the terminal phosphate group of 5′ end [33], did not interact with THGP. Based on the mechanism we found, it is expected that THGP may show antiviral activity against other RNA viruses that contain 5′ triphosphate modification in their genomes, such as vesicular stomatitis virus (VSV) and sendai virus (SenV) [34,35]. We think that it is an important future issue to explore the range of viruses that are affected by THGP. Currently, neuraminidase (NA) inhibitors such as oseltamivir and zanamivir, or endonuclease inhibitor such as baloxavir marboxil, are used as antiviral drugs for the treatment of IAV [36]. In this regard, our current data may provide a therapeutic potential of THGP as a novel type of antiviral agent that directly target the IAV genome, interfering with the interaction between the viral RNA and the IAV polymerase. 

On the other hand, given our findings on the suppressive effect of THGP on RIG-I signaling, it might also be a new therapeutic agent for patients with inflammatory diseases due to some types of RIG-I hyperactivation. Tricho-hepato-enteric syndrome (THES), a rare autosomal recessive disorder, is caused by mutations in the TTC37 or SKIV2L gene. The superkiller viralicidic activity 2-like (SKIV2L) RNA helicase, one of causative genes for tricho-hepato-enteric syndrome (THES), also known as syndromic diarrhea (SD) or phenotypic diarrhea (PD), is known to be involved in the formation of an RNA exosome that inhibits ER stress, such as the thapsigargin-induced RIG-I signaling pathway [37]. It has been reported that humans with deficiency in SKIV2L have a type I IFN signature in their peripheral blood [37], suggesting a possible therapeutic application of THGP for the treatment of such inflammatory diseases. We have an interesting observation that THGP exhibits a suppressive effect on IFN response in response to thapsigargin treatment in a mouse macrophage cell line RAW264.7 (Tadano. S, Sato. S, and Takaoka. A; unpublished data), suggesting a possible therapeutic application of THGP for the treatment of such inflammatory diseases.

In addition to this, our data also provide a potential for THGP in excessive inflammation during viral infection (Figure 1, Figure 2 and Appendix A). It has been reported that complications or, ultimately, death arising from viral infections such as severe influenza are often associated with the hyperactivation of a proinflammatory cytokine response, which is known as a “cytokine storm” [38]. THGP, which was shown to have a dual activity on the suppression of the RIG-I pathway and direct antiviral effect, might also be promising in such a clinical setting. Our present study provides a novel aspect of THGP as a potentially attractive clinical option for therapy against infectious diseases caused by not only IAV but also possibly other viruses, as well as inflammatory diseases.

## Figures and Tables

**Figure 1 viruses-13-01674-f001:**
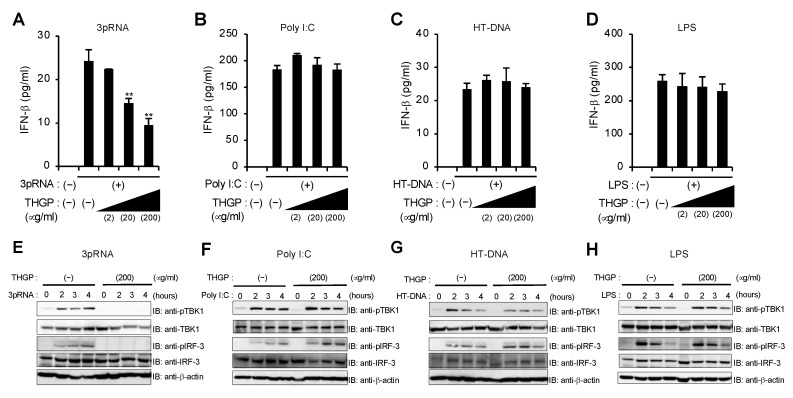
THGP suppresses IFN-β induction in response to 3pRNA but not poly I:C, HT-DNA, and LPS. (**A**–**D**) ELISA of IFN-β levels at 24 h after stimulation with 3pRNA (**A**), polyI:C (**B**), HT-DNA (**C**), and LPS (**D**) in RAW264.7 cells pretreated with indicated concentrations of THGP for 24 h. Data are presented as mean and s.d. (n = 3). (**E**–**H**) Whole cell lysates at the indicated time after stimulation with 3pRNA (**E**), polyI:C (**F**), HT-DNA (**G**), and LPS (**H**) in RAW264.7 cells pretreated with THGP for 24 h, which were prepared and subjected to immunoblotting with anti-pTBK1, TBK1, pIRF-3, IRF-3, and β-actin antibodies. ** *p* < 0.01 vs. control. Data are representative of at least three independent experiments.

**Figure 2 viruses-13-01674-f002:**
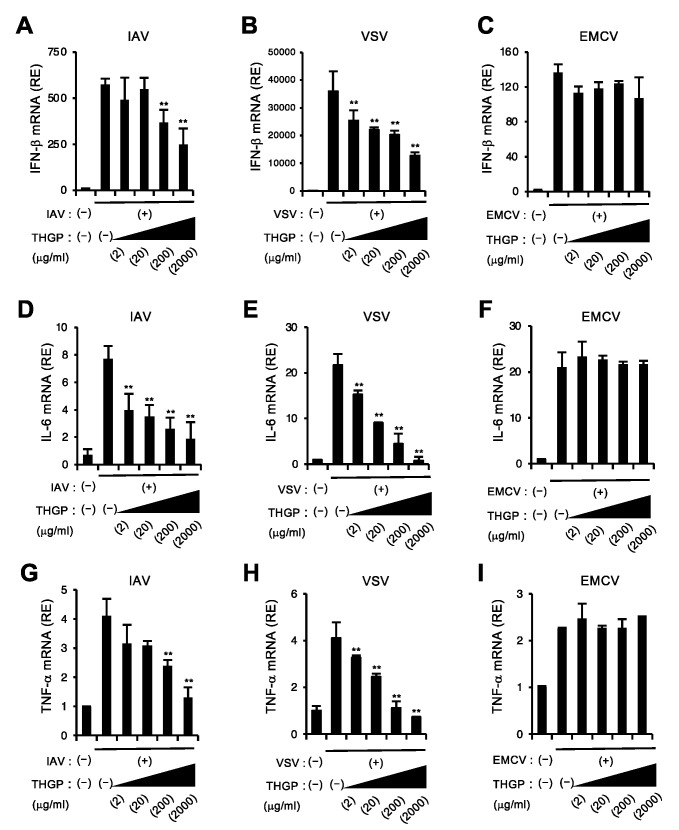
THGP reduces IFN-β induction in response to IAV and VSV but not EMCV infections. (**A**–**I**) qRT-PCR analysis of IFN-β (**A**–**C**), IL-6 (**D**–**F**), and TNFα (**G**–**I**) mRNA levels at 8 h after infection with control or IAV (**A**,**D**,**G**), VSV (**B**,**E**,**H**), and EMCV (**C**,**F**,**I**) in RAW264.7 cells pretreated with the indicated concentrations of THGP for 24 h. ** *p* < 0.01 vs. control. Data are presented as mean and s.d. (n = 3) and are representative of at least three independent experiments.

**Figure 3 viruses-13-01674-f003:**
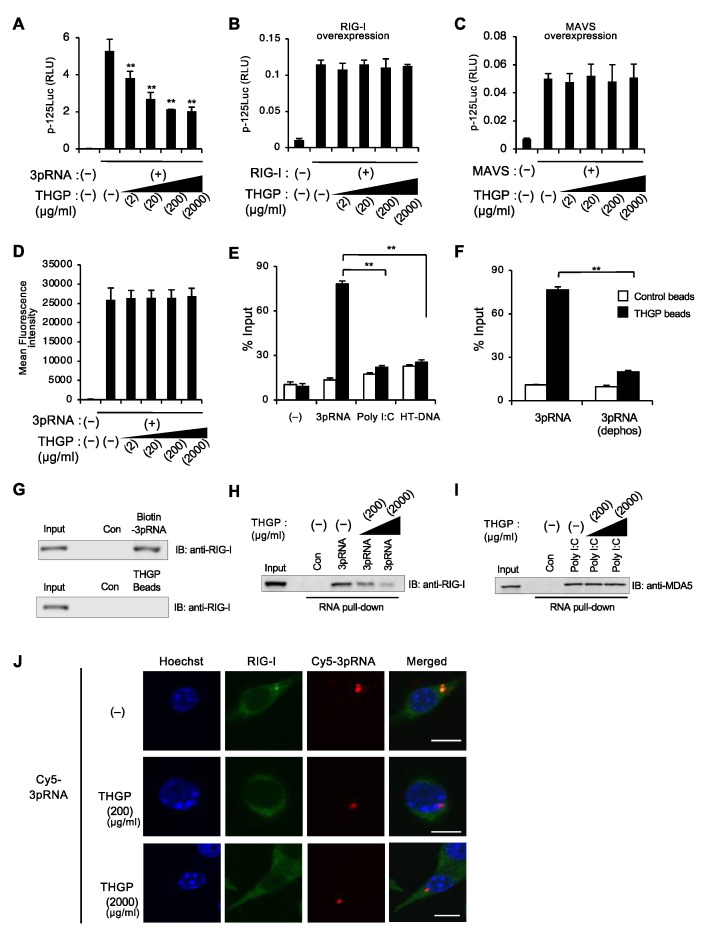
THGP interacts with 3pRNA but not Poly I:C and HT-DNA and inhibits the binding of RIG-I with 3pRNA. (**A**,**B**) Luciferase assay of IFN-β gene promoter after treatment of the indicated concentrations of THGP following the transfection of 3pRNA (**A**), RIG-I (**B**), and MAVS (**C**) in HEK293T cells. (**D**) FACS analysis at 2 h after transfection with Cy5-3pRNA in RAW264.7 cells. (**E**) THGP beads pull down assay of 3pRNA, poly I:C, and HT-DNA. The amount of precipitated RNA/DNA with THGP beads (filled bar) or control beads (opened bar) is shown. (**F**) THGP beads pull down assay of 3pRNA and 3pRNA treated with alkaline phosphatase. The amount of precipitated RNA/DNA with THGP beads (filled bar) or control beads (opened bar) is shown. (**G**) Pull-down assay showing the binding of biotinylated 3pRNA to RIG-I (top) and THGP to RIG-I (bottom). (**H**,**I**) 3pRNA (**H**) and polyI:C (**I**) pull down assay, which test the interaction of biotinylated 3pRNA and RIG-I and biotinylated polyI:C and MDA-5 in RAW264.7 cells. Co-precipitated proteins with biotinylated RNA in the presence of indicated concentrations of THGP were subjected to Western blotting using anti-RIG-I and anti-MDA-5 antibodies. (**J**) Immunofluorescence analysis for the co-localization of RIG-I and Cy5-3pRNA in the presence of the indicated concentrations of THGP. A representative of more than 30 captured cells is presented. Bar: 10 μm. ** *p* < 0.01 vs. control. Data are presented as mean and s.d. (n = 3) and are representative of at least three independent experiments.

**Figure 4 viruses-13-01674-f004:**
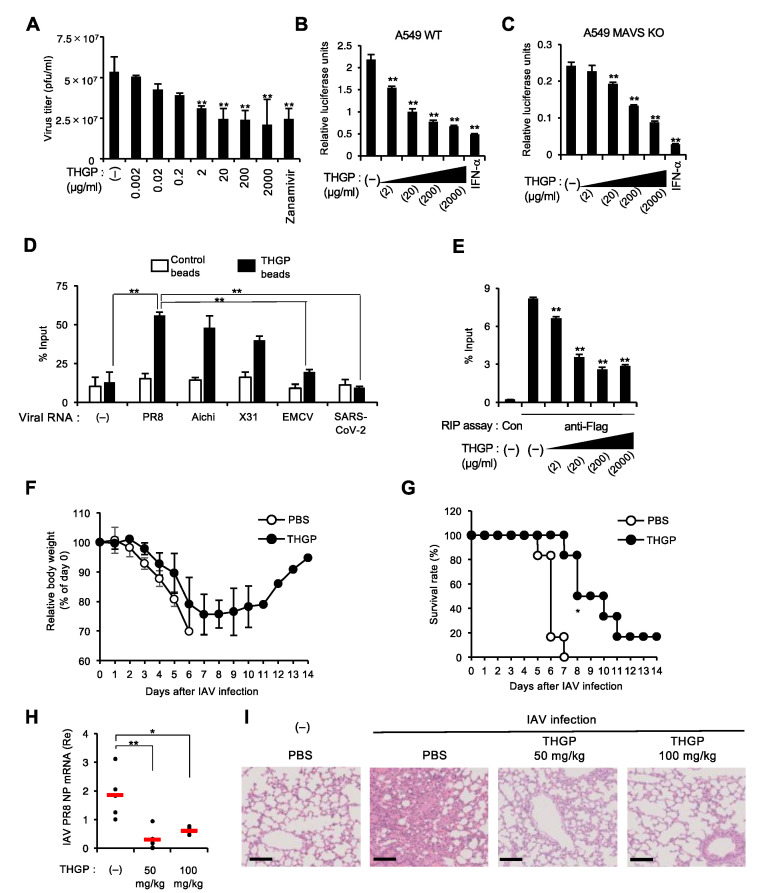
THGP directly abolishes IAV replication in vitro and in vivo. (**A**) Viral titers measured in MDCK cells after 24 h of infection with IAV in RAW264.7 cells in the presence of the indicated concentrations of THGP or 0.3 mM zanamivir as control. pfu, plaque-forming units. n = 3 samples per group. (**B**–**C**) Luciferase activity-based mini-genome assay of IAV replication in the presence of the indicated concentrations of THGP or 1000 U of IFN-α for 24 h in A549 WT (**B**) and A549 MAVS KO cells (**C**) after 24 h of transfection with IAV-related plasmids, including pPolI/ NP(0)luc2(0), Flag-PB2, PB1, PA, and NP. The CC_50_ of THGP on A549 cells is evaluated (Appendix A). (**D**) THGP pull down assay of viral RNAs including IAV PR8 (A/Puerto Rico/8/1934 H1N1), IAV Aichi (A/Aichi/2/68 H3N2), IAV X31 (H3N2), EMCV, and SARS-CoV-2, which were isolated from purified viruses. The % input of precipitated viral RNA with THGP beads is shown. (**E**) RIP assay with A549 MAVS KO cell lysates prepared after 48 h of transfection with the IAV-related plasmids including pPolI/ NP(0)luc2(0), Flag-PB2, PB1, PA, NP by using anti-Flag or control immunoglobulin G (con). The immunoprecipitated IAV RNA-like RNA of the reporter gene was measured by qRT-PCR with specific primers for firefly luciferase. (**F**–**G**) At 3 h after intranasal administration with THGP or PBS at a dose of 50 mg/kg of body weight, C57BL/6J mice were infected intranasally with 10^5^ pfu/animal of IAV PR8. Body weight change (**F**) and survival rate (**G**) were monitored (n = 6 per group). Mice were intranasally administrated with THGP or PBS at a dose of 50 mg/kg of body weight every 2 days for 14 days. Day 0 indicates the time of initiation of administration. * *p* < 0.05 vs. control (a log rank-test). We also tested the effect of THGP on the body weight and survival of uninfected WT mice (Appendix A). (**H**) qRT-PCR analysis of IAV NP RNA levels in the lung tissues of MAVS KO mice at 48 h post-infection of IAV following administration with PBS (−), THGP (50 mg/kg), or THGP (100 mg/kg). (**I**) H&E stain of lung tissues at 72-h post-infection in MAVS KO mice following i.n. administration with PBS (−), THGP (50 mg/kg), or THGP (100 mg/kg). Bar: 100 μm. ** *p* < 0.01 vs. control. * *p* < 0.05 vs. control. Data are presented as mean and s.d. (n = 3 in (A-E); n = 5 in (H)) and are representative of at least three independent experiments.

## Data Availability

The data presented in this study are available on request from the corresponding author.

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
