# Peer review of "Dual Effect of Organogermanium Compound THGP on RIG-I-Mediated Viral Sensing and Viral Replication during Influenza a Virus Infection"

_viruses, 2021, doi:10.3390/v13091674_

Round 1
Reviewer 1 Report
The work "Dual effect of organogermanium compound THGP on RIG-I- 2 mediated viral sensing and viral replication during influenza A 3 virus infection" by Sunanda Baidya et al is a well-executed study providing new data on a possible application of a hydrolysate of Ge-132, 32 3-(trihydroxygermyl) propanoic acid (THGP) as an antiviral agent. I have only a few questions, concerning the animal experiments:
Q1. Why does the intranasal route of THGP administration was chosen for mice treatment? Are there any therapeutic effects of THGP after oral or parenteral administration?
Q2. What was the injected volume of THGP solution in mice experiments?
Q3. It is not clear if the mice were sedated or not during the viral challenge and THGP instillation.
Minor correction:
Line 445: ...viruses, which contain 5’ triphosphated modification in their in their genomes such as...
Reviewer 2 Report
The authors studied the effect of THGP on influenza virus, VSV and EMCV as well as its modulation effect on immune response. They find out that THGP can downregulate type I interferon and pro-inflammatory cytokine production in the presence of 5'-triphospate or influenza virus. They hypothesise that THGP can bind RNA, which encode RNA-depend-RNA polymerase proteins of influenza virus resulted in inhibition of influenza virus replication. Finally, they find out that THGP can bind 5'-triphosphate moiety on influenza virus and VSV and can be used as anti-virotic agents against these viruses. While it was an interesting study, authors make same over-arching conclusion about mechanism of THGP as anti-viral agents. I have some comments and questions to improve the study.
- The interferons are produced to inhibit virus replication and spread. Usually, the high amounts of the IFN resulted in decreased viral titer. In this case, decreased level of IFN correlated with decreased replication of the virus. Can you explain that?
- Influenza virus activated RIG-1 as well as MDA-5. MDA-5 plays important role in induction of IRF3, NF-κB and production of cytokines
- Did authors tested the vRNA polymerase encoded PB1, PB2, PA, and NP separately? Did you test also other vRNA encoded HA, NA, NS1 etc? Was the binding of THGP specific only to RNA- dependent -RNA polymerase vRNAs?
- In the case, that the binding of THGP was specific to vRNA- dependent -RNA polymerase RNAs, how you will explain that THGP can be used against vRNA of VSV (discussion).
- H&E staining of the lung tissue is missing in Materials and Methods.
- 3 B, C and Fig.4B, C – distinguish in axes (graph) what is different.
- It is not necessary to include NS into the graphs. Everybody understand that it is not significant.
- Author should used one name for the interferons. They are using Ifn g, Ifn b, IFN-β etc. Choose one. IFN-β, IFN-γ etc. are most commonly used.
- Using IL 6 instead Il6 will be clearer, especially in the graphs.
Round 2
Reviewer 2 Report
I have no other comments.